# Efficient regression with deep neural networks: how many datapoints do we need?

**Daniel Lengyel**
Department of Computing
Imperial College London
London, United Kingdom
d.lengyel@imperial.ac.uk

**Anastasia Borovykh**
Department of Mathematics
Imperial College London
London, United Kingdom
a.borovykh@imperial.ac.uk

## Abstract

While large datasets facilitate the learning of a robust representation of the data manifold, the ability to obtain similar performance over small datasets is clearly computationally advantageous. This work considers deep neural networks for regression and aims to better understand how to select datapoints to minimize the neural network training time; a particular focus is on gaining insight into the structure and amount of datapoints needed to learn a robust function representation and how the training time varies for deep and wide architectures.

## 1 Introduction

Deep neural networks are able to achieve excellent performance when trained on big datasets, even if the input data size is very large (*e.g.* $28 \times 28$ pixels for the MNIST dataset). One explanation for their good performance in high dimensions is that the data lies in a latent space with much lower dimensionality than that of the full data encoding space. Large datasets [1], potentially in combination with the implicit bias that occurs during training [5, 23, 3], can then be seen as a reasonable way of learning robust representations.

The size of the dataset however clearly impacts the training efficiency. System constraints such as on-device computing or simply the fact that large datasets are not always available [25] has led to a renewed interest in learning over small datasets. A challenge with using small datasets is that it is not clear that the critical characteristics of a function will be captured. While augmenting a small dataset is an option when a large datasets is not available [30], other approaches [10, 34, 33, 6, 27] have focused on defining and obtaining 'good' datapoints for which the learning can be efficient and lead to a robust representation. Also in the literature on interpolators and mesh optimization schemes interesting approaches for choosing relevant datapoints have been presented [19, 17, 26, 11]. Typically, this amounts to selecting points in 'critical' areas of the function based on *e.g.* the curvature.

In this work we consider the goal of learning a function representation $\hat{f}$ from datapoints $(\mathbf{x}_i, f(\mathbf{x}_i))_{i=1}^N$ and take a first step towards answering two intimately linked questions: i) can we identify properties of good datasets?; ii) how much efficiency can be gained when learning over these good datasets compared to *e.g.* randomly selected points and how does this differ for deep and wide neural networks? The questions underlie a more fundamental understanding about the structure that data occurring in nature must have to enable computationally feasible learning (see also [41, 37]). To answer i) and ii) we will use several established data selection methods used in *e.g.* computer graphics and compare the performance of deep neural networks over these optimally sampled points and over uniform and randomly selected points. In this work the focus lies on dimensions $d = 1$ and $d = 2$.

Has it Trained Yet? Workshop at the Conference on Neural Information Processing Systems (NeurIPS 2022).

**Related work**   We briefly remark on several lines of related work. i) Prior work has shown that a DNNs good performance even in overparameterized, noisy settings can be attributed to the implicit bias [12, 5, 24]; the dimensionality seems to be crucial in minimizing the impact of noise [23, 3]. ii) In computer graphics literature the definition of an optimal approximating mesh of a given complexity is used for the efficient but realistic display of the geometric models [11, 16, 35]; the choice of those mesh points is closely related to choosing optimal datapoints and we will use their insights in this work. iii) A large line of work focuses on dataset pruning, active learning or optimal feature selection using a variety of different scalar metrics such as gradients or information theoretic quantites [34, 33, 6, 14, 18, 39, 27, 4]. For a more detailed related work review we refer to Appendix A.

## 2   Methodology for dataset generation

We discuss the methods we will use for constructing training sets of 'good' datapoints.

**Curvature-based sampling for one dimension**   We discuss the method used for optimal point selection for $d = 1$. The curvature is frequently used as a metric to determine the critical points of a surface approximation. The work of [26] reparametrizes the curve according to the curvature. Given a function $y = f(x)$, the curvature is given by $k(x) = \frac{f''(x)}{(1+f'(x)^2)^{3/2}}$. Observe that, if the first-order gradient is small, the curvature is approximated by the second-order derivative highlighting the relationship between curvature and Hessian. The goal is then to define a density so that sampling from this density leads to samples from high-curvature regions. Define, $f_k(x) = \int_{x_0}^{x} k(u)du$, with $x_0$ the beginning of the domain[1]. This defines a cumulative density-like function, where high curvature areas will align with a steeper $f_k(x)$ as a function of $x$. In order to sample from the density underlying this function, we then use the inverse transform sampling method. Sample $U \sim \mathcal{U}[a,b]$; then define $T(U) = (f_k)^{-1}(U)$. Then, $X \sim f_k$, *i.e.* it will be sampled in high curvature regions.

**Mesh optimization for two dimensions**   For $d = 2$ we will use the iterative mesh simplification method as used in [11]. This method defines a triangulation: a mesh that 'covers' a surface using simplices[2]. The vertices of the mesh lie on the function itself and these vertices are connected through simplex faces (*i.e.* a generalization of piecewise linear approximations to arbitrary dimension). It assumes an initial triangulation is given which consists of an arbitrary number of vertices. The goal of the algorithm is to simplify this triangulation by simplifying the number of edges and vertices, to obtain a triangulation with a *given* number of vertices. It does so by iteratively contracting sets of vertices if the cost associated to their contraction (*i.e.* how much accuracy is lost by removing one of the vertices in the approximated surface) is not too high. As shown in [17, 11] at convergence the resulting mesh has sampled more points in regions of high curvature. This shows the value of curvature and gradients in general: it arises even in approximation algorithms that do not directly optimize for the curvature but instead minimize some loss between the mesh and the original surface. For completion we include in Appendix C more details on the mesh optimization scheme.

## 3   Numerical results

Our goals are: i) understand how 'optimally' sampled points influence the model performance, ii) understand how 'optimally' sampled points can speed up convergence, iii) obtain results on the number of points needed to learn a robust function.

**Experimental setup**   We report results for the Dixon-Price function, the Michalewicz function and a modified Rastrigin function (see Appendix D). We want to ensure that all methods are able to learn from the data (*i.e.* that they contain the critical function information). The training data is thus generatred using the following methods: i) a uniform deterministic grid defined over the domain, ii) uniform randomly sampled grid over the domain, iii) for one-dimension we use the curvature reparamterization from Section 2, iv) for two dimensions we use the mesh simplification algorithm from Section 2. We denote with $N_W$ the number of nodes per layer and $N_L$ the number of layers. We

---

[1]We remark that the work of [26] uses $f_k(x) = \int_{x_0}^{x} k(u)||f'(u)||du$; this comes from wanting the integral to be invariant to parametrizations in the curve parameter which in our case is not needed.

[2]In one dimension it amounts to an optimal piecewise linear approximation.

report the mean squared error (MSE) for one dimension over 500 datapoints and for two dimensions $200^2$ datapoints on a uniform grid. We use or $1e4$ training epochs and a learning rate of $1e-3$.

### 3.1 One dimension

We compare the curvature-based sampling method against uniform (random) grids for $d = 1$.

**Rastrigin function** We present the results for $N = 12$, $d = 1$ and a neural network architecture of $N_W = 32$ and $N_L = 2$ trained with Adam. Figure 1 and Tables 1 - 2 show a clear benefit in using the curvature-sampled datapoints. The number of training iterations required to reach a certain test performance is smaller when using 'optimal' datapoints than when using a uniform grid. Interestingly, when the neural network has fewer parameters the difference between the two sampling strategies becomes smaller, showing that the combination of overparametrisation and implicit bias allows the neural network to reach much better performance on 'optimal' datapoints.

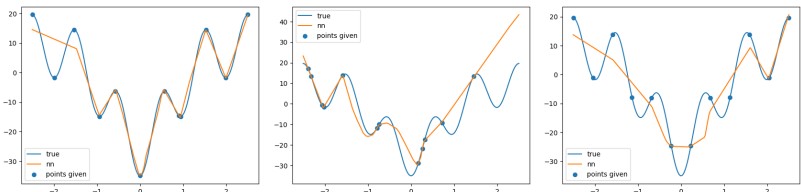

Figure 1: Rastrigin function for $d = 1$. Left to right: curvature-based sampling (MSE is 16.85), uniform random sampling (MSE is 137.88 ), uniform grid (MSE is 44.53)

Table 1: Rastrigin function. (L) Effect of model architecture: (train, test) performance averaged over 5 runs. (R) Effect of number of datapoints for arch. (32,2): test performance.

| $(N_W, N_L)$ | Uniform Grid | Curvature | Nr of datapoints | Uniform Grid | Curvature |
|---|---|---|---|---|---|
| (8,2) | (1.97,48.2) | (1.87,41.6) | 10 | 52.1 | 59.0 |
| (32,2) | (1.26,38.6) | (0.88,24.8) | 12 | 36.0 | 48.9 |
| (8,4) | (1.59,39.2) | (0.65,32.6) | 20 | 30.8 | 18.1 |

Table 2: Rastrigin function. Effect of number of training iterations: test performance averaged over 5 runs.

| Number of iterations | Uniform Grid | Curvature |
|---|---|---|
| 500 | 53.8 | 49.2 |
| 1000 | 50.1 | 44.2 |
| 2000 | 48.0 | 36.6 |
| 3000 | 47.2 | 35.1 |

**Michalewicz function in one dimension** Figure 2 shows that the curvature outperforms the random grid for the Michalewicz function.

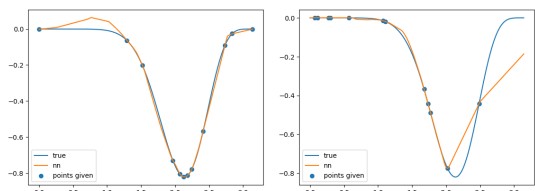

Figure 2: Michalewicz function for $d = 1$. Left to right: curvature-based sampling (MSE is 0.00070), uniform random sampling (MSE is 0.010).

**Dixon function in one dimension** As the results in Figure 3 show, the curvature is not always the ideal metric as the uniform grid in certain cases may outperform the curvature-sampled points.

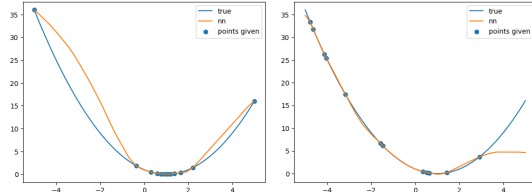

Figure 3: Dixon function for $d = 1$. Left to right: curvature-based sampling (MSE is 16.85), uniform random sampling (MSE is 5.17).

## 3.2 Higher dimensions

We now compare the performance of deep and wide neural networks for the mesh simplification method against the uniform grid for $N_W = 256$ and $N_L = 2$. Figure 4 shows the error for the Rastrigin function where 'optimal' points outperform the random grid. However, it also show that also in $d = 2$ the optimal datapoint selection does not work for the Michalewicz function. We conclude that in $d = 2$ mesh optimization *may* have benefits in speeding up convergence but there is room for improving the datapoint sampling method.

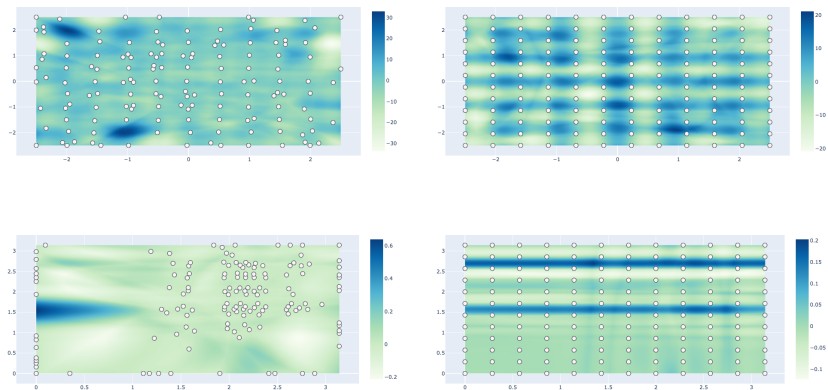

Figure 4: (T) Error for the Rastrigin function for $d = 2$. Left to right: mesh optimization-based sampling (MSE is 44.9), uniform grid (MSE is 48.3). (B) Error for the Michalewicz function for $d = 2$. Left to right: mesh optimization-based sampling (MSE is 0.010), uniform grid (MSE is 0.0030).

## 4 Discussion

In this work we showed how optimally choosing datapoints *could* result in more efficient convergence of the neural network models to robust function representations; interestingly, optimal datapoints seem to work best in combination with significant overparametrisation. However, as was shown, the used metrics and algorithms (curvature and mesh simplification) for certain function structures did not outperform the uniform grid, especially in higher dimensions. One promising direction is to use repulsive particles to spread them across the function space in a way that balances distance and high-curvature points could prove beneficial [22]. We hope to address this with better data generation strategies in future work. We also to extend this work to functions in higher dimensions to better understand how the number of needed datapoints depends on dimensionality and in what dimensionality it remains computationally feasible to learn. A recent line of work *e.g.* [31, 36] focuses on solving partial differential equation (PDE) solutions through neural networks; whether the optimal point selection can be used in that setting is to be investigated.

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

## A    Details on the related work

**Interpolation and implicit bias**    It has been shown that neural networks in an overparametrized regime are able to *interpolate*, *i.e.* perfectly fit, the data, while still achieving good generalization performance contrary to conventional wisdom which warns that when fitting noisy data interpolation may lead to overfitting. In part, the neural network's good performance can be attributed to the implicit bias [12, 5, 24]: even if the network has a high capacity, the model structure in combination with the training method leads to the learned function to be sufficiently regularized in between the datapoints. This implicit bias can be strong enough to minimize the impact of noise if the parameter space is sufficiently *high-dimensional* [23, 3]. The work of [21] compares interpolation against regression and similarly shows high-dimensional setups with noisy data where interpolation performs better than regression technique. However, [8] highlights how the implicit bias should not be too strong when noise is present. Somewhat contrary to the idea that good performance can be achieved in interpolating regimes, the work of [9] shows that even interpolation on noise-free data may hurt generalization to out-of-distribution datasets. The extent to which the implicit bias performs in small, even noise-free, data settings is however still an open question. While we are not directly focusing on extrapolation (defined as the performance outside of the convex hull of the datapoints) in this work, we remark on an interesting line of work that studies the extrapolation ability of neural networks *e.g.* [2, 40].

**Mesh optimization and piecewise linear approximators**    Given a set of datapoints, various interpolation methodologies exist, *e.g.* polynomial functions [7], splines [13], piecewise linear functions [19] and Delauney triangulation [35]; the linear methods can be seen as defining a certain approximating structure of linear pieces (a tesselation). The linear interpolators are frequently used in computer graphics where the aim is to obtain an optimal approximating mesh of a given complexity for the efficient but realistic display of the geometric models [11, 16, 35]. For these methods it is often known how to select the datapoints so that the interpolating function has a small error on the whole of the domain. These methods typically consist of an iterative simplification of the mesh by applying sequentially a set of mesh transformation steps *e.g.* edge collapse which unifies vertices into one [22, 17] or sample points according to some metric that describes which points are critical, *e.g.* the curvature [26]. The nonconvex nature of deep learning results in it being unclear if the methods used for datapoint selection in other interpolators will work for the deep learning setup. We highlight the distinction between methods that represent the surface solely through points [22, 28, 26] and those that combine point selection with the optimal interpolation *i.e.* the geometric structure of the mesh [17, 16, 11]. We also remark on the line of research that defines piecewise linear approximators; crucial to these setups is to find the values of the points that connect the linear segments [19].

**Dataset pruning and feature selection** Given a big dataset, to improve computational efficiency one may want to decrease its size. Pruning the dataset is one technique to achieve this. Various different metrics have been proposed based on *e.g.* uncertainties or distances to other samples [34, 33, 6, 14, 18, 39], norms of the gradient [27], information theoretic quantities [4] such as mutual information [29] or entropy [15, 20]; or other work uses optimization-based approaches [42] or genetic algorithms [38]. In this work we take a slightly different approach and use established results from linear interpolation and unlike the previously mentioned papers that focus on classification our work focuses on the less-studied regression setup.

# B   Background on polynomials and their error bound

For a complete reference on polynomial approximations and interpolation we refer the reader to [7]. Consider a polynomial $p_n(x) = a_0 + a_1 x + \ldots + a_n x^n$. It is well-known that given $n + 1$ distinct datapoints $x_0, ..., x_n$ and $n + 1$ values $f_0, ..., f_n$, there exists a unique polynomial $p_n$ for which $p_n(x_i) = f(x_i)$ for $i = 0, ..., n$: The following result for a polynomial approximation is well-known.

**Theorem B.1** (Unique polynomial representation; Thm 2.1.1 [7]). *Given $n + 1$ distinct points $x_0, ..., x_n$ and $n + 1$ values $f_0, ..., f_n$, there exists a unique polynomial $p_n(x) \in \mathcal{P}^n$ (space of $n$-th order polynomials) for which $p_n(x_i) = f_i$, $i = 0, ..., n$.*

*Proof.* The proof follows from applying Cramer's rule and noting that the determinant of the system is non-zero for distinct points. $\square$

Remainder theory is able to quantify the accuracy of the approximation. The upper bound in (4) can be obtained through the following theorem.

**Theorem B.2** (Remainder for $n$th order polynomia; Thm 3.1.1 in [7]). *Consider $f(x) \in C^n[a, b]$ such that $f^{(n+1)}(x)$ exists for $x \in [a, b]$. Consider points $a \leq x_0 < x_1 < ... < x_n \leq b$. Then the following holds,*

$$f(x) - p_n(x) = \frac{(x - x_0) \ldots (x - x_n)}{(n + 1)!} f^{(n+1)}(\xi), \tag{1}$$

*with $\min(x, x_0, ..., x_n) < \xi < \max(x, x_0, ..., x_n)$.*

*Proof.* Define the functions,

$$K(x) = \frac{f(x) - p_n(x)}{(x - x_0) \ldots (x - x_n)}, \tag{2}$$

$$W(t) = f(t) - p_n(t) - (t - x_0) \ldots (x - x_n) K(x). \tag{3}$$

Note that $W(t)$ vanishes at $x_0, x_1, ..., x_n, x$. By Rolle's theorem $W^{(n+1)}$ must vanish at $\xi$ given by $\min(x, x_0, ..., x_n) < \xi < \max(x, x_0, ..., x_n)$. Since $W^{(n+1)}(t) = f^{(n+1)}(t) - (n + 1)!K(x)$ and using $W^{(n+1)}(\xi) = 0$ one obtains $K(x) = \frac{1}{(n+1)!} f^{(n+1)}(\xi)$ and plugging this into the definition of $K(x)$ the statement follows. $\square$

The error of the polynomial approximation can be upper bounded as follows (see corollary 3.1.3 in [7]):

$$|f(x) - p_n(x)| \leq \frac{|x - x_0| \ldots |x - x_n|}{(n + 1)!} \max_{a \leq t \leq b} |f^{(n+1)}(t)|. \tag{4}$$

The term $\max_{a \leq t \leq b} |f^{(n+1)}(t)|$ is independent of the *choice* of interpolation points; to minimize the error through an *optimal* choice of datapoints $(x_i, f_i)$, $i = 0, ..., n$ one would thus focus on minimizing $\max_{a \leq x \leq b} |(x - x_0) \ldots (x - x_n)|$. The optimal points are given by the zeros of the Tschebysheff polynomials. The Tschebysheff polynomials of order $n$ are defined as follows.

**Definition 1** (Tschebysheff polynomials). *The Tschebysheff polynomial of degree $n$ is given by,*

$$T_n(x) = x^n + \binom{n}{2} x^{n-2}(x^2 - 1) + \binom{n}{4} x^{n-4}(x^2 - 1)^2 \cdot (n = 0, 1, ...) \tag{5}$$

The zeros of the Tschebysheff polynomials are given by the following theorem.

**Theorem B.3** (Zeros of Tschebysheff polynomials; Theorem 3.3.3 in [7]). $T_n(x)$ *has simple zeros at the $n$ points, $x_k = \cos\frac{2k-1}{2n}\pi$, $k = 1, 2, ..., n$.*

A linear approximation is obtained by setting $n = 1$ in the polynomial $p_n$. From (4) it follows that for a linear approximation the error is bounded by the Hessian of the function $f$. This highlights the role of higher-order gradients in optimal point selection. The work of [19] extends the linear approximation to a *piecewise* linear approximation for convex functions. Using (4) it becomes clear that in order to obtain the optimal piecewise linear approximation one needs to find the linear segments such that the maximum Hessian over each those segments is minimal.

## C  Details on the mesh optimization method

Denote with $(\mathcal{K}, \mathcal{V})$ a triangulation, where $\mathcal{K}$ is a simplicial complex specifying the connectivity of the vertices, edges and faces; $\mathcal{V} = (\mathbf{v}_1, ...)$ with $\mathbf{v}_i = [v_i^x, v_i^y, v_i^z, 1]^T$ is a set of vertex positions. We assume we are given an initial triangulation; this initial configuration can consist of an arbitrary amount of vertices $\mathbf{v}_i$, $i = 1, ...$ on the surface. The algorithm proceeds by iteratively contracting a set of vertex pairs $(\mathbf{v}_1, \mathbf{v}_2) \rightarrow \bar{\mathbf{v}}$. Only valid pairs are considered for contraction, where the validity of a pair is defined by i) $(\mathbf{v}_1, \mathbf{v}_2)$ is an edge, or ii) $||\mathbf{v}_1 - \mathbf{v}_2|| \leq \epsilon$, where the latter allows for the re-joining of disjoint triangles. Each contraction is associated with a cost $\Delta(\mathbf{v}) = \mathbf{v}^T Q \mathbf{v}$, where $Q$ is a $4 \times 4$ symmetric matrix. The matrix $Q$ is defined as follows: using that each vertex is associated with a set of triangles that can be seen as planes $\mathbf{p} = [a, b, c, d]^T$ defined through $ax + by + cz + d = 0$, the error is computed as the summed error over the planes, the 'distance-to-plane' metric [32]:

$$\Delta(\mathbf{v}) = \sum_{\mathbf{p} \in \text{Planes}} (\mathbf{p}^T \mathbf{v})T = \mathbf{v}^T \sum_{\mathbf{p} \in \text{Planes}} K_p \mathbf{v}, \tag{6}$$

where $K_p = \mathbf{p}\mathbf{p}^T$. The location of $\bar{\mathbf{v}}$ is consequently chosen as the one that optimizes $\Delta(\bar{\mathbf{v}})$, and since the error function is quadratic, finding its minimum is done by solving a linear problem. The final algorithm is then given as follows: i) compute $Q$, ii) select valid pairs, iii) compute for each valid pair $(\mathbf{v}_1, \mathbf{v}_2)$ the optimal contraction vertex $\bar{\mathbf{v}}$ and error $\bar{\mathbf{v}}^T(Q_1 + Q_2)\bar{\mathbf{v}}$ (where $Q_1$ and $Q_2$ are the quadrics at $\mathbf{v}_1$ and $\mathbf{v_2}$, iv) remove the minimum cost vertex, v) repeat until the required number of vertices is reached. The vertices and the function value at the vertices will be used as the train dataset.

We remark that many alternatives for the greedy iteration over the loss function exist; *e.g.* the methods of [17, 16] that maintain meshes that are optimal with respect to an appearance/loss metric or works like [22] aim to directly optimize the vertices by considering them as particles evolving via some repelling forces.

## D  Details on the experimental setup

Let $\mathbf{x} \in \mathbb{R}^d$. In our setup $d = 1, 2$. The Michalewicz function is given by,

$$f(\mathbf{x}) = -\sum_{i=1}^{d} \sin(x_i) \sin^{2m}\left(\frac{i x_i^2}{\pi}\right), \tag{7}$$

with $m = 2$. The Dixon-Price function is given by,

$$f(\mathbf{x}) = (x_1 - 1)^2 + \sum_{i=1}^{d} i(2x_i^2 - x_{i-1})^2. \tag{8}$$

The modified Rastrigin function is given by,

$$f(\mathbf{x}) = -25e^{-\frac{||\mathbf{x}||^2}{1.5}} - 10e^{\frac{||\mathbf{x}||^2}{0.5}} + 10d + \sum_{i=1}^{d} -10\cos(2\pi x_i). \tag{9}$$

Here we have removed the quadratic term and replaced it with a sum of squared exponentials.

