# OpenReview forum: "Efficient regression with deep neural networks: how many datapoints do we need?"
_NeurIPS.cc/2022/Workshop/HITY — HITY Workshop NeurIPS 2022_

### Official Review · Reviewer_7hHt · 2022-10-17
**Accept: Interesting study of data point selection methods’ impact on the performance of neural network regression models**

**Rating:** 1
**Confidence:** 3

**Review:**


The paper studies how training a neural network regression model on subsets of the training data, which are chosen by different selection strategies, impacts its predictive performance.

I recommend to accept the paper, since it begins with the exploration of an interesting research direction.

---

### Official Review · Reviewer_WCkD · 2022-10-18
**A study that applies deep learning for regression on small samples of synthetic data**

**Rating:** 1
**Confidence:** 3

**Review:**

This paper focuses on dataset pruning/active learning strategies for regression using deep learning (DL). Specifically, it tackles the important questions of whether we can identify properties of good (sub-)datasets for regression and whether we can quantify the benefit of training DL models on good data.
In order to do that, the authors propose an experimental setup in which DL models are trained on 2 kinds of samples of 1D and 2D synthetic data: one using informed sampling algorithms from the literature (curvature-based for 1D and meshes for 2D), against using random sampling. Results show that informed samples have a mixed impact on model performance, convergence speed and generalization ability.

One core assumption in the paper is that "good" datapoints are associated with high degrees of curvature. From an empirical point of view, the experiments presented are at very small scale and the functions generated seem somewhat arbitrary. Still, the work is principled and offers a new perspective on the topic that may be worth exploring further (see my coments below), especially given that mixed results were obtained in a context where good results would be intuitively expected, hinting at potential discoveries. For that reason I propose to accept the paper.

Questions/Comments:
1. In this paper it is assumed that "optimally" selected datapoints are the ones that present high curvature on the data domain itself (e.g. edges on an image). While this sounds intuitively reasonable, it would be interesting to add a compelling reason as to why this is the case for any general regression problem. Otherwise the goals in 3 are difficult to characterize in a systematic way, and a limited number of low-dimensional experiments may replace this characterization (particularly when, as stated in line 36, "the dimensionality seems to be crucial in minimizing the impact of noise"). Some references that could help with this are the Universal Approximation Theorem and Unser's Representer Theorem, which tackle the ability of Neural Networks to interpolate arbitrary data  using splines (see https://link.springer.com/article/10.1007/s10208-020-09472-x).
2. To the question of how to quantify the benefits of learning on "good data", it is worth mentioning that theoretical work has been developed to prove that weakly labeled data is in the same efficiency category as fully labeled data (e.g. see Chris Re's lab publications at SAIL https://arxiv.org/abs/1711.10160). Maybe some of those techniques can serve as inspiration to characterize the meaning of "good datapoints" for generic regression problems.
3. On the empirical side: One technique that could allow the extension of the techniques presented here to higher-dimensional, more realistic data (at least for the mesh-based method) is parametric UMAP, which is a scalable and topologically principled way of learning/approximating and projecting a manifold: https://umap-learn.readthedocs.io/en/latest/parametric_umap.html
4. Regarding the relation between dataset size and model size, work on deep linear models may be of help: https://arxiv.org/abs/1710.03667

---

### Decision · Program_Chairs · 2022-10-20

Accept